# Robust Identifiability in Linear Structural Equation Models of Causal Inference

**Karthik Abinav Sankararaman**[1]     **Anand Louis**[2]     **Navin Goyal**[3]

[1]Meta AI , Austin, USA,
[2]Indian Institute of Sciences, Bengaluru, India,
[3]Microsoft Research, Bengaluru, India,

## Abstract

We consider the problem of robust parameter estimation from observational data in the context of linear structural equation models (LSEMs). Under various conditions on LSEMs and the model parameters the prior work provides efficient algorithms to recover the parameters. However, these results are often about generic identifiability. In practice, generic identifiability is not sufficient and we need robust identifiability: small changes in the observational data should not affect the parameters by a huge amount. Robust identifiability has received far less attention and remains poorly understood. Sankararaman et al. (2019) recently provided a set of sufficient conditions on parameters under which robust identifiability is feasible. However, a limitation of their work is that their results only apply to a small sub-class of LSEMs, called "bow-free paths." In this work, we show that for *any* "bow-free model", in all but $\frac{1}{\text{poly}(n)}$-measure of instances *robust identifiability* holds. Moreover, whenever an instance is robustly identifiable, the algorithm proposed in Foygel et al., (2012) can be used to recover the parameters in a robust fashion. In contrast, for generic identifiability Foygel et al., (2012) proved that with measure 1, instances are generically identifiable. Thus, we show that robust identifiability is a *strictly* harder problem than generic identifiability. Finally, we validate our results on both simulated and real-world datasets.

## 1 INTRODUCTION

Causal inference is a central problem in a variety of fields in the natural and social sciences. The goal of causal inference is to design methodologies that infer if a group of events *cause* a particular phenomenon or not. A canonical

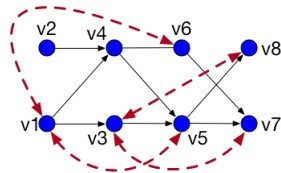

Figure 1: Illustration of a 2-bow-free graph where the maximum in-degree and out-degree in any vertex is 2. Black solid lines represent causal edges and red dotted lines represent correlation of the noise parameters.

example is the age-old debate on whether smoking causes cancer (Terry [1964]). The causal inference problem has been extensively studied in statistics, economics, epidemiology, computer science among others (*e.g.,* Guyon et al. [2010], Holland et al. [1985], Pearl [2009], Peters et al. [2017], Pearl and Mackenzie [2018]) and several schools of thought exist. One important and popular model is the *linear structural equation* model (LSEM); see, *e.g.,* Bentler and Weeks [1980] and Bollen [1989]. Informally, the experimenter has a model of the world and a dataset (represented as samples from a latent distribution) collected during the experiment. The goal is to use the samples and the model to infer the strength of dependencies between various quantities of interest. In LSEM, the experimenter's model is a Gaussian linear model which is formally defined as follows.

The model of the causal relationship is given by a mixed graph $G = (V, E, F)$, where the vertex set $V$ of size $n$ corresponds to the set of observable random variables. Let $\mathbf{X} \in \mathbb{R}^{n \times 1}$ denote the vector of random variables corresponding to the vertices in $V$. The set $E$ of *directed* edges captures the direction of causality in the model: an edge from vertex $u$ to vertex $v$ implies that $\mathbf{X}_u$ causes $\mathbf{X}_v$. We will assume that the edges in $E$ form an acyclic directed graph. The set $F$ of *bidirected* edges denotes the presence of confounding effects (described shortly). Let $\eta \in \mathbb{R}^{n \times 1}$ denote a vector of noise random variables whose covariance matrix is given by $\mathbf{\Omega} \in \mathbb{R}^{n \times n}$. We assume that $\eta$ is a zero-mean multivariate Gaussian random variable. Let

*Accepted for the 38th Conference on Uncertainty in Artificial Intelligence* (UAI 2022).

$\mathbf{\Lambda} \in \mathbb{R}^{n \times n}$ denote the matrix of edge weights on the directed edges; the entries in $\mathbf{\Lambda}$ can be interpreted as encoding the strength of causal influence.

The LSEM model posits that the dependencies between observed variables are linear: the effect on a particular random variable $\mathbf{X}_u$ is jointly determined by its immediate parents in the directed component of the graph plus a Gaussian noise $(\eta_u)$, which we can represent as

$$\mathbf{X} = \mathbf{\Lambda}^T \mathbf{X} + \eta. \tag{1}$$

The edge set $E$ puts constraint on the zero pattern of $\mathbf{\Lambda}$: if $(u, v) \notin E$, then $\mathbf{\Lambda}_{(u,v)} = 0$. Let us denote the set of such matrices by $W(E)$. The bidirected edge set $F$ specifies the zero pattern of $\mathbf{\Omega}$: if $u \neq v$ and $(u, v) \notin F$, then $\mathbf{\Omega}_{(u,v)} = 0$. Let $PD(F)$ denote the set of positive semidefinite matrices satisfying this constraint, and let $PD$ be the set of positive semidefinite matrices whose dimensions will be clear from the context. We assume that the dataset is sampled from a distribution that is unknown to the experimenter and has the following properties.

Since the random vector $\eta$ is a Gaussian random variable with mean zero, it follows that $\mathbf{X}$ is also a Gaussian random variable with mean zero. Thus, the tuple $(\mathbf{\Lambda}, \mathbf{\Omega})$ defines the distribution of $\mathbf{X}$. We are interested in this map and its invertibility. Since $\mathbf{X}$ is Gaussian, instead of working with its distribution we can work with its covariance matrix which is a sufficient statistic. This is what we will do in the sequel. Let $\mathbf{\Sigma}$ denote the covariance matrix of $\mathbf{X}$ and let $\Phi_G : (\mathbf{\Lambda}, \mathbf{\Omega}) \mapsto \mathbf{\Sigma}$ be the map of interest. From the linear relationship in Eq. (1) we have

$$\mathbf{\Sigma} = (\mathbf{I} - \mathbf{\Lambda})^{-T} \mathbf{\Omega} (\mathbf{I} - \mathbf{\Lambda})^{-1}. \tag{2}$$

Hence, the map $\Phi_G : W(E) \times PD(F) \to PD$ is given by

$$\Phi_G : (\mathbf{\Lambda}, \mathbf{\Omega}) \mapsto (\mathbf{I} - \mathbf{\Lambda})^{-T} \mathbf{\Omega} (\mathbf{I} - \mathbf{\Lambda})^{-1}.$$

The (global) identifiability question for $G$, namely are the parameters $(\mathbf{\Lambda}, \mathbf{\Omega})$ recoverable from $\mathbf{\Sigma}$ for all $\mathbf{\Sigma} \in PD$, has a positive answer iff $\Phi_G$ is invertible. The class of mixed graphs $G$ for which $\Phi_G$ is invertible has been precisely characterized by Drton et al. [2011]. But this turns out to be too strong a restriction and a slightly weaker notion of *generic identifiability* is considered. A mixed graph $G$ is said to be generically identifiable if for almost all $(\mathbf{\Lambda}, \mathbf{\Omega}) \in W(E) \times PD(F)$, we can recover these parameters from $\Phi_G(\mathbf{\Lambda}, \mathbf{\Omega})$. Here "almost all" is meant in the measure-theoretic sense for any reasonable measure such as the Lebesgue or Gaussian measure on $W(E) \times PD(F)$.

A central question in the study of LSEMs is determining if a mixed graph is *generically identifiable* (GI) and estimating the parameters from the covariance matrix when GI does hold. While mixed graphs for which GI holds have not been completely characterized, many classes of such graphs have

been found, (*e.g.,* Brito and Pearl [2006], Drton and Weihs [2016], Drton et al. [2011], Foygel et al. [2012], McDonald [2002]). In particular, *bow-free* graphs (Brito and Pearl [2006]) form one such class and will be studied in this paper. For this class, we can first compute the matrix $\mathbf{\Lambda}$ from the covariance matrix $\mathbf{\Sigma}$ and then recover $\mathbf{\Omega}$ by computing $(\mathbf{I} - \mathbf{\Lambda})^T \mathbf{\Sigma} (\mathbf{I} - \mathbf{\Lambda})$. Since this does not involve matrix inversion, this can be done in a robust manner. Note that this $\mathbf{\Omega}$ may not satisfy the zero-patterns mandated by the model; however, this can be remedied by solving the convex optimization problem for finding the closest PSD matrix satisfying the required zero-pattern. Triangle inequality implies that the optimal solution to the convex optimization problem is a PSD matrix that is also close to the original $\mathbf{\Omega}$ with the same zero-patterns. Thus, we will be primarily interested in the inverse map

$$\Psi_G^{-1} : \mathbf{\Sigma} \to \mathbf{\Lambda}. \tag{3}$$

Much of the prior work has focused on designing algorithms with the assumption that the *exact* joint distribution over the variables is available, which in turn gives exact $\mathbf{\Sigma}$. However, in practice, the data is noisy and inaccurate and the joint distribution is generated via *finitely many* samples from this noisy data. This leads to the question of (generic) *robust identifiability* (RI): if $\mathbf{\Sigma}$ is perturbed slightly, does $\Psi_G^{-1}(\mathbf{\Sigma})$ change only slightly? We will formalize this notion in terms of the condition number. For parameter estimation algorithms to be useful we need robust identifiability to hold because of unavoidable inaccuracies in the input in practice.[1] Motivated by this, the key question we consider in this paper is the following.

*Are bow-free LSEMs robustly identifiable?*

**Our contributions and discussion.** We answer the question in affirmative, by showing that the space of instances for which the identifiability algorithm in Foygel et al. [2012] is robust is *large*. In other words, for a natural measure over the space of parameters $(\mathbf{\Lambda}, \mathbf{\Omega})$ for acyclic graphs satisfying *bow-free* condition, the probability that an instance can be robustly identified using the algorithm in Foygel et al. [2012] is at least $1 - \frac{1}{\text{poly}(n)}$, where $n$ denotes the number of observable variables in the system. This is in contrast to generic identifiability, where the authors in Foygel et al. [2012] show that the probability is $1$. To achieve this, we prove a stronger statement, namely, sufficient conditions for robust identifiability an arbitrary instance should satisfy when perturbed with adversarial noise (See Fig. 2 where random instances violating it can lead to exponential growth of condition number). Then we show that when the instances

---

[1] In fact, Schulman and Srivastava [2016] and Sankararaman et al. [2019] construct families of examples where the inaccuracies compound to lead to a large error in the final output in semi-Markovian models and LSEMs respectively.

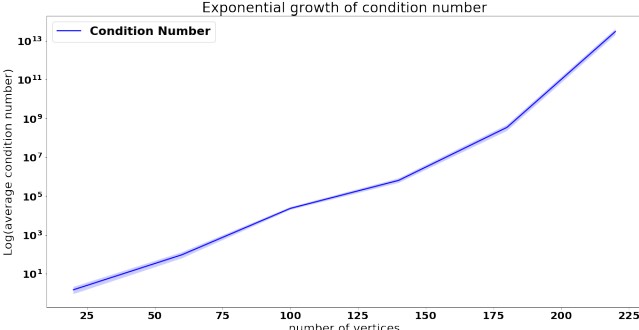

Figure 2: Randomly generated instance that is ill-conditioned when it violates Assumption (A.3) in Model 1. We generate a fixed graph that is layered (*i.e.,* edge in topological order $i$ only goes to that in topological order $i + 1$) with max-degree of $4$. For every directed edge $i \rightarrow j$ we have $\lambda_{i,j} \sim \mathcal{U}[-1.2, 1.2]$ and $\mathbf{\Omega}$ is randomly generated according to Model 2.

are sampled from a natural measure over the set of instances (*i.e.,* uniform distribution over $\mathbf{\Lambda}$ and Wishart distribution over $\mathbf{\Omega}$), it satisfies the sufficient condition with probability at least $1 - \frac{1}{\text{poly}(n)}$. We corroborate our theoretical analysis with simulations on a gene expression dataset used in Drton et al. [2009] and also on additional simulated datasets. Our paper has both conceptual and technical novelty compared to Sankararaman et al. [2019]. First, Sankararaman et al. [2019] analyze the error accumulated on every edge; such a strategy fails for anything beyond paths. Here, we instead analyze the total error accumulated across many edges together. The key challenge is in finding the right set of edges to be grouped. Here we show that we need to analyze the total error in computing the weight parameter of all the incoming edges to a vertex $v$. On the technical side, while we use the same high-level idea of induction, we need to work with matrices instead of scalars. This brings up many new non-trivial challenges requiring matrix-theoretic arguments.

It is occasionally pointed out that the algorithms mentioned above (*e.g.,* Foygel et al. [2012]) are designed for the purpose of identifiability and not for parameter estimation, and as such should not be used for the latter. While, a priori, this could be true, for the specific case of the above algorithms we do not see any reason for not using them for parameter estimation other than the fact that they assume access to the exact covariance matrix. That the access to the exact covariance matrix is not essential under reasonable conditions on parameters is in fact the main point of our paper. This shows that the algorithms designed assuming exact access *can* be used for parameter estimation in realistic situations. It's also pertinent to note here that the field of robust statistics seeks to deal with similar situations (under various models of perturbations, often adversarial) by designing new algorithms with the explicit goal of robust identifiability (see Diakonikolas and Kane [2019] and references therin for a

recent survey). Our results show that under a reasonable model of perturbation, existing algorithms are already robust. We are not aware of any work on LSEMs in the robust statistics literature.

A related point is that if one were to not use the above algorithms for parameter estimation then one needs alternative algorithms. Unfortunately, we are not aware of any algorithms with provable guarantees for parameter estimation other than the ones mentioned above—regardless of the access to the covariance matrix being exact or not. RICF algorithm (Drton et al. [2009]) is designed expressly for parameter estimation using the maximum likelihood principle from finitely many samples. Maximum likelihood based algorithms come equipped with confidence intervals which provide an estimate of uncertainty in parameter estimation and could potentially be useful for our problem. Unfortunately this is not the case: For one, we are not aware of a quantitative analysis using confidence intervals. Second, we allow adversarial perturbations for which confidence intervals are not applicable. Third, while practically useful, RICF does not provide any theoretical guarantees on finding the correct parameters. It only guarantees that the parameters it finds achieve a local maximum of the likelihood (there are empirical indications that under some conditions it does find the global maximum). Thus, there is a need for algorithms for parameter estimation with provable guarantees without assuming exact access to the covariance matrix or the distribution. As already mentioned, in this paper we show that the existing identifiability algorithms are in fact such algorithms under reasonable conditions on parameters. For another discussion of the identifiability vs. estimation issue we refer the reader to a recent manuscript (Maclaren and Nicholson [2019]), though they do not provide any positive result like ours.

**Related work.** The issue of robust identifiability for causal models has started to gain attention only recently. Schulman and Srivastava [2016], Sankararaman et al. [2019], Maclaren and Nicholson [2019], Gordon et al. [2021] are the only papers we are aware of. Schulman and Srivastava [2016] showed by means of an example that the recovered parameters can be very sensitive to errors in the data and so robust recovery is not always possible. They worked in the setting of semi-Markovian models (see, *e.g.,* Shpitser and Pearl [2008]). Their example is carefully constructed for the purpose of showing that robust recovery is not possible, and it is not clear if such examples are likely to arise in practice. In other words, their result leaves open the possibility that robust recovery may be possible for a large part of parameter space (according to some reasonable probability measure). A result in this direction was provided by Sankararaman et al. [2019] for a subclass of LSEMs. For bow-free paths they show that if the parameters are chosen from a certain random distribution then the parameters are robustly identifiable with high probability. Our results in

the present paper build upon Sankararaman et al. [2019]. In particular, our Lemma 1 generalizes Lemma 1 in Sankararaman et al. [2019]. Moreover, given this lemma, the proof for the bound on the condition number follows as in prior work. Finally, Maclaren and Nicholson [2019] provide an abstract framework for studying the robust identifiability problems within the context of causal inference. They also relate it to the extensive literature on similar problems in statistics and inverse problems and provide an entry point to this literature.

Ghoshal and Honorio [2018, 2017] gave an algorithm for parameter estimation and structure learning for linear SEMs from observational data with theoretically good sample and computational complexity and under stochastic noise under certain conditions on the parameters. However, they make the strong assumption that the noise covariance matrix $\mathbf{\Omega}$ is diagonal (and in the second paper under the stronger assumption that $\mathbf{\Omega}$ is a multiple of the identity) which may be overly restrictive in many settings (Drton et al. [2009]). Thus their result is not comparable to ours.

There is also a significant body of work on problems such as model misspecification. These are related to but are distinct from the problem studied in the present paper. We refer to [Sankararaman et al., 2019, Sec. 1.2] for references and commentary on the differences. A very recent example in the same vein is (Cinelli et al. [2019]). Again, while sharing similar general motivation, this work is complementary to ours.

## 2 PRELIMINARIES

**Notation.** Throughout this paper, we use the notation $G = (V, E, F)$ to represent a causal mixed graph structure where $V$ denotes the set of vertices, $E$ denotes the set of directed (causal) edges and $F$ denotes the set of bidirected (covariance of noise) edges. For simplicity, we assume that the vertices in the set $V$ are indexed $\{1, 2, \ldots, |V|\}$. Throughout this paper, we assume that the directed edges $E$ induce an acyclic graph. For a matrix $\mathbf{A}$, we use the notation $\|\mathbf{A}\| := \max_{\mathbf{x} \neq 0} \frac{\|\mathbf{Ax}\|_2}{\|\mathbf{x}\|_2}$ to denote the spectral norm of this matrix. For a vector $\mathbf{b}$, we denote $\|\mathbf{b}\| = \sqrt{\mathbf{b}^T \cdot \mathbf{b}}$ to be the 2-norm. We use many standard properties of the spectral norm in the proofs of this paper. Lemma 7 in the appendix summarizes these for completeness. We use $\sigma_1(\mathbf{A}), \lambda_1(\mathbf{A})$ to denote the largest singular and eigenvalue respectively, of matrix $\mathbf{A}$. We let $\mathbf{\Lambda}_{[I, J]}, \mathbf{\Omega}_{[I, J]}, \mathbf{\Sigma}_{[I, J]} \in \mathbb{R}^{|I| \times |J|}$ to denote the sub-matrix of $\mathbf{\Lambda}, \mathbf{\Omega}, \mathbf{\Sigma}$ respectively, corresponding to vertices in the index set $I$ and $J$. For two given vertices $u, v \in V$, we use $\mathbf{\Lambda}_{u,v}, \mathbf{\Omega}_{u,v}, \mathbf{\Sigma}_{u,v}$ to refer to the $(u, v)$-entry of respective matrices. We use $\mathrm{poly}(n)$ to denote a function which is polynomial in $n$. $\mathcal{U}[a, b]$ denotes the uniform distribution on the interval $[a, b]$ with pdf $f(x) = 1/(b - a)$.

We denote $\mathtt{layer}(i)$ to be the set of vertices such that $v \in \mathtt{layer}(i)$ if and only if the longest directed path ending in $v$ has length $i$. Thus, $\mathtt{layer}(1)$ denotes the set of vertices with no incoming directed edge. For any vertex $v$, we denote $\mathrm{pa}(v)$ to be the set of vertices in $V$ such that there is a directed edge from every vertex in $\mathrm{pa}(v)$ to $v$. Additionally, we use the notation $\mathrm{spa}(v) := \mathrm{pa}(\mathrm{pa}(v))$. Since the graph is acyclic, there exists a topological sort order of the vertices $V$ (Foygel et al. [2012]). Throughout this paper, we assume that $n$ is an asymptotic parameter; thus $o(1)$ denotes terms that go to $0$ as $n$ goes to infinity.

**Definition 1** ($k$-bow-free causal graphs). *A causal graph $G = (V, E, F)$ is called a $k$-bow-free causal graph if it has the following properties.*

1. ***Bow-free.*** *The graph is bow-free* i.e., *between any two vertices $u$ and $v$, there is never both a directed and bidirected edge.*

2. ***Maximum in-degree or out-degree of $k$.*** *For any vertex $v \in V$, the total number of directed edges coming into $v$ is at most $k$. Likewise, the total number of directed edges leaving $v$ is also $k$. Thus, $|\mathrm{pa}(v)| \leq k$ for every $v \in V$.*

Figure 1 pictorially denotes an example of $k$-bow-free causal graph. Throughout this paper, $k$ should be viewed as a small constant (for instance in our experiments $k$ is either 2 or 7). As in prior work (Sankararaman et al. [2019], Schulman and Srivastava [2016]), we use the notion of *condition number* to measure the robustness of the models. Before we define the condition number, we define the relative distance between two matrices. Given matrices $\mathbf{A}, \mathbf{B} \in \mathbb{R}^{n \times m}$, we define the *relative distance*, denoted by $\mathrm{Rel}(\mathbf{A}, \mathbf{B})$ as the following: $\mathrm{Rel}(\mathbf{A}, \mathbf{B}) := \max_{\substack{1 \leq i \leq n, \\ 1 \leq j \leq m: \\ |A_{i,j}| \neq 0}} \frac{|A_{i,j} - B_{i,j}|}{|A_{i,j}|}$. The $\ell_\infty$-condition number is defined as follows.

**Definition 2** (Relative $\ell_\infty$-condition number). *Let $\mathbf{\Sigma}$ be a given data covariance matrix and $\mathbf{\Lambda}$ be the corresponding parameter matrix. Let a $\gamma$-perturbed family of matrices be denoted by $\mathcal{F}_\gamma$ (i.e., set of matrices $\tilde{\mathbf{\Sigma}}_\gamma$ such that $\mathrm{Rel}(\mathbf{\Sigma}, \tilde{\mathbf{\Sigma}}_\gamma) \leq \gamma$). For any $\tilde{\mathbf{\Sigma}}_\gamma \in \mathcal{F}_\gamma$ let the corresponding recovered parameter matrix be denoted by $\tilde{\mathbf{\Lambda}}_\gamma$. Then the relative $\ell_\infty$-condition number is defined as,*

$$\kappa(\mathbf{\Lambda}, \mathbf{\Sigma}) := \sup\nolimits_{\gamma < \frac{1}{n^4}} \mathtt{ess\ sup}_{\tilde{\mathbf{\Sigma}}_\gamma \in \mathcal{F}_\gamma} \frac{\mathrm{Rel}(\mathbf{\Lambda}, \tilde{\mathbf{\Lambda}}_\gamma)}{\mathrm{Rel}(\mathbf{\Sigma}, \tilde{\mathbf{\Sigma}}_\gamma)}. \quad (4)$$

Condition number as the notion of stability is useful since a bound on this quantity translates to an upper-bound on the sample complexity. More precisely, to get an error of $\epsilon$ in the output polynomial in $1/\epsilon$, condition number and other parameters of the input number of samples suffice (*e.g.,* Srivastava et al. [2013]).

## 2.1 REQUIRED BACKGROUND FROM PRIOR WORK

We give a self-contained background needed from Foygel et al. [2012] for our paper.

**Definition 3** (Half-trek (Foygel et al. [2012])). *For any given vertex $v \in V$, the set $htr(v)$ denotes the set of vertices that can be reached from $v$ via a path of the form,*

$$v \leftrightarrow v_1 \rightarrow v_2 \rightarrow \ldots \rightarrow v_d \quad or, v \rightarrow v_2 \rightarrow \ldots \rightarrow v_d.$$

**Definition 4** (Parameter Recovery Algorithm from Foygel et al. [2012]). *Consider a vertex $v \in V$ such that $pa(pa(v)) \neq \phi$. The goal is to compute the vector $\mathbf{\Lambda}_{[pa(v), \{v\}]}$. Let $Y_v = \{y_1, y_2, \ldots, y_k\}$ be a given set of vertices corresponding to vertex $v$. Let $pa(v) = \{p_1, p_2, \ldots, p_k\}$ denote the set of parents of $v$. Let $\mathbf{A}$ be a matrix such that $A_{i,j} = [(\mathbf{I} - \mathbf{\Lambda})^T \cdot \mathbf{\Sigma}]_{y_i, p_j}$ if $y_i \in htr(v)$ and $A_{i,j} = [\mathbf{\Sigma}]_{y_i, p_j}$ otherwise. Likewise, let $\mathbf{b}$ denote a vector such that $b_i = [(\mathbf{I} - \mathbf{\Lambda})^T \cdot \mathbf{\Sigma}]_{y_i, v}$ if $y_i \in htv(v)$ and $b_i = [\mathbf{\Sigma}]_{y_i, v}$ otherwise. Then we have,*

$$\mathbf{\Lambda}_{[pa(v), v]} = \mathbf{A}^{-1} \cdot \mathbf{b}. \tag{5}$$

*For vertices $v \in V$ such that $pa(pa(v)) = \phi$ we compute $\mathbf{\Lambda}_{[pa(v), \{v\}]}$ using the expression,*

$$\mathbf{\Lambda}_{[pa(v), v]} = \mathbf{\Sigma}_{[pa(v), pa(v)]}^{-1} \cdot \mathbf{\Sigma}_{[pa(v), v]}. \tag{6}$$

# 3 INVERSE PROBLEM WITH ADVERSARIAL NOISE

In this section, we consider LSEMs with $k$-bow-free graph and show that under a sufficient condition (formally defined in the assumptions of Model 1), these models can be robustly identified using the algorithm in Foygel et al. [2012] in the presence of *adversarial* noise. The model we consider is as follows.

**Model 1.** *We consider the following model of perturbation. Assume that we are given a data covariance matrix $\mathbf{\Sigma}$. Let $\mathcal{E} \in \mathbb{R}^{n \times n}$ denote the matrix of perturbations. Fix a small $0 < \gamma < \frac{1}{n^4}$. Thus, the perturbed matrix is $\tilde{\mathbf{\Sigma}} := \mathbf{\Sigma} + \mathcal{E}$. Additionally, we posit the following property on the perturbation. For every entry $(i, j)$ we have $\mathcal{E}_{i,j} \leq \frac{\gamma}{\sqrt{k}} \Sigma_{i,j}$. WLOG we assume that there exists an entry $(i, j)$ such that $\mathcal{E}_{i,j} = \frac{\gamma}{\sqrt{k}} \Sigma_{i,j}$. We have the following assumptions for every vertex $v \in V$.*

**(A.1)** *__Input Condition Number.__ The condition number of the principal sub-matrix $\mathbf{\Sigma}_{[pa(v), pa(v)]}$, defined as $\kappa(\mathbf{\Sigma}_{[pa(v), pa(v)]}) := \|\mathbf{\Sigma}_{[pa(v), pa(v)]}^{-1}\| \|\mathbf{\Sigma}_{[pa(v), pa(v)]}\| \leq \kappa_0 \leq \frac{1}{2\gamma}$.*

**(A.2)** *__Diagonal dominance.__ For some $0 < \alpha < 1$, the following hold:*

$$\|\mathbf{\Sigma}_{[pa(v), v]}\| \leq \alpha \|\mathbf{\Sigma}_{[pa(v), pa(v)]}\|,$$
$$\|\mathbf{\Sigma}_{[spa(v), pa(v)]}\| \leq \alpha \|\mathbf{\Sigma}_{[pa(v), pa(v)]}\| \ and$$
$$\|\mathbf{\Sigma}_{[spa(v), v]}\| \leq \alpha \|\mathbf{\Sigma}_{[pa(v), pa(v)]}\|.$$

**(A.3)** *__Normalized parameters.__ We have $\|\mathbf{\Lambda}_{[spa(v), pa(v)]}\| \leq \beta < 1$. Additionally, for every directed edge $(u \rightarrow v)$ in the causal DAG, we have $|\Lambda_{u,v}| \geq \frac{1}{\lambda} > \frac{1}{n^2}$ where $\Lambda_{u,v}$ represents the edge-weight.*

**Intuition on the assumptions.** Before we state our theorem, we provide some intuition on the assumptions. An upper-bound on the condition number of the input matrix (as in Assumption (A.1)) is a necessary condition even in the simplest case of robustly solving a system of linear equations. More specifically, the relative error in solving a system of linear equations compared to a perturbed instance is upper-bounded by the condition number of the constraint matrix (Example 3.4 in Stewart [1998]). Since LSEMs significantly generalize this, it is natural that such a condition should be *necessary*. Assumption (A.3) states that $\|\mathbf{\Lambda}\|$ corresponding to all incoming edges for any set of vertices $pa(v)$ is upper-bounded by a constant less than 1. Intuitively, it means that the total "information" passed from the vertices appearing earlier in the topological order to those in the later parts does not blow up. The *a priori* limiting assumption is Assumption (A.2); this is required for technical reasons to make the analysis go through. Intuitively, this assumption is a version of the *diagonal-dominance* in matrices; however, we require a comparison between a principal sub-matrix and a neighboring $k$-dimensional sub-matrix. We show in Section 4 that under an arguably natural generative model for LSEMs, with high probability the generated LSEM satisfies Assumption (A.2) suggesting that it is in fact not a strong assumption.

The main result of the paper is the following bound on the $\ell_\infty$-condition number of any bow-free LSEM satisfying the assumptions in Model 1.

**Theorem 1.** *Consider a $k$-bow-free causal model denoted by the mixed graph $G = (V, E, F)$. If $\alpha\beta\kappa_0 < 0.99$ and $\frac{\alpha\kappa_0}{1-\alpha\beta\kappa_0}\left(1 + \frac{\kappa_0(1+\beta)}{1-\alpha\beta\kappa_0}\right) < \frac{0.99}{k}$ then for the model of perturbations described in Model 1 we have that the condition number $\kappa(\mathbf{\Lambda}, \mathbf{\Sigma}) \leq \mathcal{O}\left(\frac{n^2}{\sqrt{k}}\right)$.*

To prove the main theorem, we first show the following lemma which bounds the difference between the true and the recovered parameter.

**Lemma 1.** *If $\alpha\beta\kappa_0 < 1$ and $\frac{\alpha\kappa_0}{1-\alpha\beta\kappa_0}\left(1 + \frac{\kappa_0(1+\beta)}{1-\alpha\beta\kappa_0}\right) < \frac{0.99}{k}$ then for every $v \in \mathtt{layer}(j)$ and every $j \geq 2$ we have, $\|\mathbf{\Lambda}_{[pa(v), v]} - \tilde{\mathbf{\Lambda}}_{[pa(v), v]}\| \leq \eta \cdot \gamma$, where $\eta$ is the following*

*depending on the parameters in Model 1.*

$$\eta := 10 * \left( \frac{\alpha \kappa_0^2 (1+\beta)(1+\beta+o(1))}{(1-\alpha\beta\kappa_0)^2} + \frac{\kappa_0 \alpha (1+\beta+o(1))}{1-\alpha\beta\kappa_0} \right) \cdot$$
$$\left( 1 - \frac{\alpha\kappa_0}{1-\alpha\beta\kappa_0} - \frac{\alpha\kappa_0^2(1+\beta)}{(1-\alpha\beta\kappa_0)^2} \right)^{-1} + o(1).$$

**Proof outline.** At a high level, our proof strategy is similar in spirit to that of Sankararaman et al. [2019]; they prove an analogous result for graphs that are paths (for a model similar to Model 1). However, since we prove such a result for general graphs, our setting faces many additional technical challenges. Similar to Sankararaman et al. [2019], we prove the main technical Lemma 1, using induction over the layers. For any vertex $v$, we can compute $\mathbf{\Lambda}_{[\text{pa}(v),\, v]}$ using equations 5 and 6. Using the induction hypothesis, we get that $\mathbf{\Lambda}$ for the previously considered layers has a sufficiently "small" error. Let $\mathbf{A}_v$ and $\mathbf{b}_v$ denote $\mathbf{A}$ and $\mathbf{b}$ from equation 5 for vertex $v$ when working with the true (unperturbed) $\Sigma$, and let $\tilde{\mathbf{A}}_v$ and $\tilde{\mathbf{b}}_v$ denote the corresponding matrices for $\tilde{\Sigma}$. We show that the spectral norm of the matrix $\tilde{\mathbf{A}}_v - \mathbf{A}_v$ and the norm of the vector $\tilde{\mathbf{b}}_v - \mathbf{b}_v$ is sufficiently small. We use this and bounds on the norms of $\mathbf{A}_v$ and $\mathbf{b}_v$ to show that the norm of $\tilde{\mathbf{A}}_v^{-1}\tilde{\mathbf{b}}_v - \mathbf{A}_v^{-1}\mathbf{b}_v$ is small. These steps pose multiple subtle technical challenges in comparison to Sankararaman et al. [2019], and require new ideas to handle them.

**Proof of Theorem 1.** From Lemma 1 we have that $\|\mathbf{\Lambda}_{[\text{pa}(v),\, v]} - \tilde{\mathbf{\Lambda}}_{[\text{pa}(v),\, v]}\| \leq \eta\gamma$. From Prop. (P.6) in Lemma 7 we have that the absolute value of every entry in the matrix $(\mathbf{\Lambda}_{[\text{pa}(v),\, v]} - \tilde{\mathbf{\Lambda}}_{[\text{pa}(v),\, v]})$ is at most $\|\mathbf{\Lambda}_{[\text{pa}(v),\, v]} - \tilde{\mathbf{\Lambda}}_{[\text{pa}(v),\, v]}\| \leq \eta\gamma$. Combining this with Assumption (A.3) we have $\text{Rel}(\mathbf{\Lambda}, \tilde{\mathbf{\Lambda}}) \leq \frac{\eta\gamma}{\lambda}$. Moreover, from Model 1 we have that $\text{Rel}(\Sigma, \tilde{\Sigma}) = \frac{\gamma}{\sqrt{k}}$. Thus, we get that the condition number is at most $\kappa(\mathbf{\Lambda}, \Sigma) \leq \frac{\eta\sqrt{k}}{\lambda}$. From Assumption (A.3), we have $\frac{1}{\lambda} \geq \frac{1}{n^2}$ which implies that $\kappa(\mathbf{\Lambda}, \Sigma) \leq \eta\sqrt{k}n^2$. From the definition of $\eta$ and the premise of Theorem 1 we have that $\eta \leq \mathcal{O}\left(\frac{1}{k}\right)$. Thus, we get the stated bound.

## 4   RANDOM MODEL PARAMETERS

In this section, we will consider LSEMs that are generated from random model parameters and show that they satisfy the model properties in Model 1. Thus, we show that on a *large* set of input parameters the assumptions in Model 1 hold with high-probability. Combining this with Theorem 1 implies that inputs from this parameter space can be robustly identified using *existing* algorithms *provably*.

**Model 2** (Generative model)**.** *Every non-zero entry in* $\mathbf{\Lambda} \in \mathbb{R}^{n \times n}$ *is an i.i.d. sample from the uniform distribution* $\mathcal{U}\left[-\frac{1}{2k\mu}, \frac{1}{2k\mu}\right] \setminus \left[-\frac{1}{n^2}, \frac{1}{n^2}\right]$ *for some fixed* $\mu \geq 10(k+1)$*.*

*The matrix* $\mathbf{\Omega} \in \mathbb{R}^{n \times n}$ *is generated as follows. We sample vectors* $\mathbf{v}_1, \mathbf{v}_2, \ldots, \mathbf{v}_n \in \mathbb{R}^d$ *from a $d$-dimensional unit sphere such that the following correlation holds. Each vector* $\mathbf{v}_i$ *is a uniform sample from the sub-space perpendicular to* $\text{SPAN}(\{\mathbf{v}_j\}_{j \in V_{I-1}})$*. The matrix* $\mathbf{\Omega}$ *is constructed by letting the $(i, j)$-th entry be* $\langle \mathbf{v}_i, \mathbf{v}_j \rangle$*. Thus, this matrix follows the zero-patterns mandated by the model.*

For the Model 2 defined above, we have the following theorem.

**Theorem 2.** *Let* $\mu \geq 10(k+1)$*,* $\alpha = \frac{1}{\mu} + o(1)$*,* $\beta = \frac{1}{\mu}$*,* $\lambda = n^2$*,* $\kappa_0 = \left(\frac{1+\mu}{\mu}\right)^4 + \frac{(\mu+1)^2}{5\mu^2(\mu-1)} + o(1)$*. Then with probability at least* $1 - \frac{1}{\text{poly}(n)}$ *the following hold simultaneously.*

1. *For every* $v \in V$ *we have* $\kappa(\Sigma_{[\text{pa}(v),\, \text{pa}(v)]}) \leq \kappa_0$*.*

2. *For every* $v \in V$*, we have that* $\|\Sigma_{[\text{pa}(v),\, v]}\| \leq \alpha\|\Sigma_{[\text{pa}(v),\, \text{pa}(v)]}\|$*,* $\|\Sigma_{[\text{spa}(v),\, \text{pa}(v)]}\| \leq \alpha\|\Sigma_{[\text{pa}(v),\, \text{pa}(v)]}\|$ *and* $\|\Sigma_{[\text{spa}(v),\, v]}\| \leq \alpha\|\Sigma_{[\text{pa}(v),\, \text{pa}(v)]}\|$*.*

3. *For every directed edge* $(u \to v)$ *in the causal DAG, we have that* $\frac{1}{n^2} \leq |\Lambda_{u,v}|$*. Moreover, for every* $v \in V$ *we have,* $\|\mathbf{\Lambda}_{[\text{spa}(v),\, \text{pa}(v)]}\| \leq \beta$*.*

**Proof Outline.** We prove high-probability bounds on the norm of sub-matrices of $\mathbf{\Omega}$ and $\mathbf{\Lambda}$ using the concentration properties of the inner-product of the random vectors. We then use the Taylor series expansion for $(\mathbf{I} - \mathbf{\Lambda})^{-1}$ to obtain an expression for $\Sigma$. Using the various properties of the spectral norm of matrices, and the computed high-probability bounds we obtain the required bounds.

## 5   EXPERIMENTS

In this section, we describe the results of our simulation studies. We first perform simulation studies to identify the importance of the various assumptions in Model 1 on *random perturbations*. Next we use both real-world and simulated datasets to study the effect of *graph sparsity* on the condition number. Thus, using simulations we complement our theoretical understanding of the problem and show evidence of good and bad conditioned instances that go well beyond the sufficient conditions proved.

### 5.1   SIMULATIONS TO UNDERSTAND THE IMPORTANCE OF EACH OF THE ASSUMPTIONS IN MODEL 1

To study the effect of various assumptions on the growth of condition number, we perform the following study. We generate $\mathbf{\Lambda}$ and $\mathbf{\Omega}$ randomly using the same generative model in Section 4. We generate random perturbations by modifying each non-zero entry of the associated $\Sigma$ using a $\mathcal{N}(0, 1e-2)$

random variable independently. We compute the condition number averaged over 20 independent runs of the perturbation. To violate the various assumptions, we vary the parameter $\beta$ that is used to determine the range $|\lambda_{i,j}|$ for a directed edge $i \to j$. Recall that since $\beta = \frac{1}{\mu}$ we also affect the values of $\alpha$ and $\kappa_0$. We look at the following scenarios and compute the growth of the condition number in each of these cases. *Our biggest take-away is that the assumption that affects the growth of condition number (*i.e., *exponential versus polynomial) is Assumption (A.3)*. Moreover, the constants proved in theory are worst-case perturbations; for random perturbations, our simulations show that these can potentially be significantly improved while maintaining robustness.

1. All the assumptions in theory are satisfied (Figure 3).

2. All assumptions except Assumption (A.3) are satisfied (Figure 4).

3. Both Assumption (A.3) and Assumption (A.2) are slightly violated (Figure 5).

4. Assumption $\alpha * \beta * \kappa_0 < 1$ and Assumption (A.3) are violated (Figure 7).

5. Assumption $\frac{\alpha \kappa_0}{1 - \alpha \beta \kappa_0} \left( 1 + \frac{\kappa_0 (1+\beta)}{1 - \alpha \beta \kappa_0} \right) < \frac{0.99}{k}$ and Assumption (A.3) are violated (Figure 6).

6. Large edge weights with $\beta > 1$ (Figure 8).

As can be seen from the figures, except in the last scenario (Figure 8), in all other scenarios the condition number does not grow exponentially.

## 5.2 EFFECT OF GRAPH SPARSITY.

We consider general *bow-free* graphs and random noise. Before we describe the experimental procedure, we briefly describe the challenges in running experiments; this explain why experiments in prior works are almost non-existent. The key issue with experimentation is that the ground-truth model is *unknown* and the datasets do not come with the true underlying model. In particular, LSEM is a model-based approach where designing the right model is part of the hypothesis held by the experimenter. The dataset only contains the observational data; part of the challenge in inferring causality using LSEM is in devising an appropriate model based on domain knowledge. Thus, here and in prior works (Drton et al. [2009], Sankararaman et al. [2019]) the experimental setup *simulates* various possible hypotheses in the hypothesis space.

**Gene expression dataset.** We use the dataset that corresponds to experiments on gene expression in *Arabidopsis thaliana* from Wille et al. [2004]. We look at the 13 genes which belong to a single pathway. There are $n = 118$ microarray experiments. Thus, the input matrix $\mathbf{X} \in \mathbb{R}^{118 \times 13}$.

We have 13 vertices, one corresponding to each of the genes. First, we choose a random permutation $\pi$ to order the vertices. For any pair of vertices $i, j$ such that $\pi(i) < \pi(j)$ we add a directed edge from $i$ to $j$ with probability $p$. For every vertex $j$, we choose a vertex $i \neq pa(j)$ uniformly at random and add a bidirected edge between $i$ and $j$. For every other pair of vertices, if there exists no directed edge between them, we add a bidirected edge with probability $0.1$. For a given value of $p$, we generate 30 random graph structures using the above procedure. To evaluate the condition number, we add independent $\mathcal{N}(0, \epsilon^2)$ noise to each entry in the matrix $\mathbf{X}$ to obtain the perturbed dataset $\tilde{\mathbf{X}}_\epsilon$. We then compute the corresponding covariance matrix $\tilde{\Sigma}_\epsilon$. We use the algorithm in Foygel et al. [2012] to recover parameters $\Lambda$ and $\tilde{\Lambda}_\epsilon$ corresponding to the matrices $\Sigma$ and $\tilde{\Sigma}_\epsilon$. For a given realization of the random graph, we generate 20 different datasets $\tilde{\mathbf{X}}_\epsilon$. For each of these 20 datasets, we compute the corresponding covariance matrices and run the parameter recovery algorithm Foygel et al. [2012] on them. We then average the condition numbers (*i.e.,* maximum relative change in $\Lambda$ to the maximum relative change in $\Sigma$) across various values realizations of the random graph. Thus, a single experiment is averaged over the 30 different random graphs multiplied by the 20 different runs for a fixed graph. We run two kinds of experiments for each $p$: (1) in which we normalize the dataset (*i.e.,* every row in the matrix $\mathbf{X}$ has a norm of 1) (2) in which the dataset is not normalized. Figure 9 shows the results of our experiments. We run simulations for $p \in \{0.05, 0.1, 0.2, 0.3, \ldots, 0.9\}$. As can be seen from the results when the values of $p$ are small (sparse regime), the average condition number tends to be small. However, as the value of $p$ becomes large (dense models) the condition number increases. This can be explained by the fact that when errors across many edges accumulate, the total error gets compounded.

**Simulated dataset.** We consider two sets of experiments that differ in the number of vertices in any layer: we consider $k = 2$ and $k = 7$. For each setting of $k$, we consider $p = 0.2$ (sparse regime) and $p = 0.8$ (dense regime). When $k = 2$ we consider graphs where the total number of vertices is in the set $\{20, 30, 40, 50\}$ while when $k = 7$ the number of vertices were in the set $\{14, 21, 35, 49\}$. For each triple $(k, p, n)$, we generate many random graphs exactly as in the main section of the experiments. We generate a random $\Lambda$ corresponding to the random graph instance, where every edge is given a weight uniformly at random from $[-\text{range}, \text{range}]$. We use two values of range in the experiments (range $= 1/7$ and range $= 1$). For every bidirected edge between $(i, j)$ we sample a $\mathcal{N}(0, 1)$ random variable $\omega$ and let both $\Omega_{i,j} = \Omega_{j,i} = \omega$. For every $i \in [n]$ we let $\Omega_{i,i}$ to be the sum of absolute values in row $i$ added to a $\chi_1^2$-random variable.[2] The construction implies that $\Omega$ is a Symmetric Diagonally Dominant matrix and thus, is

---

[2]This is the exact setup in Drton et al. [2009].

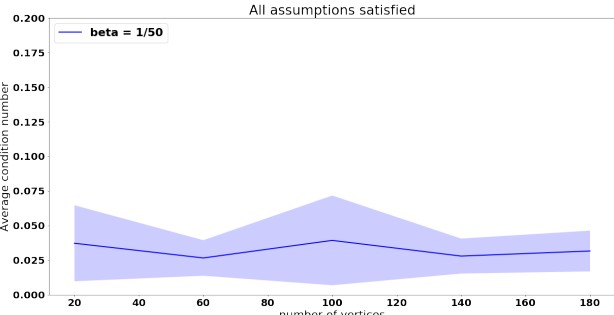

Figure 3: All assumptions satisfied

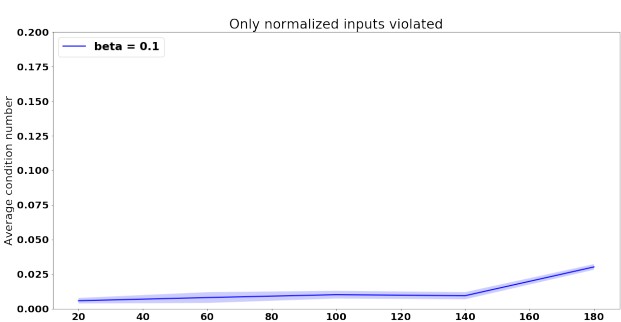

Figure 4: Only normalized input violated.

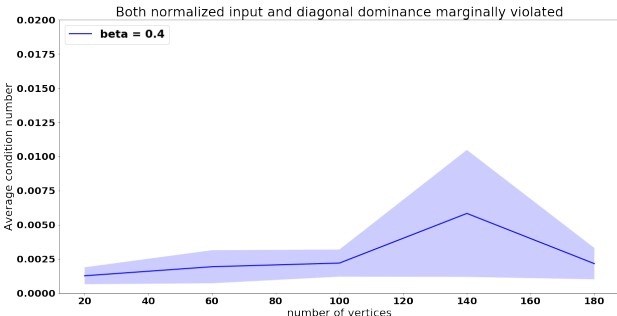

Figure 5: Normalized input and diagonal dominance marginally violated.

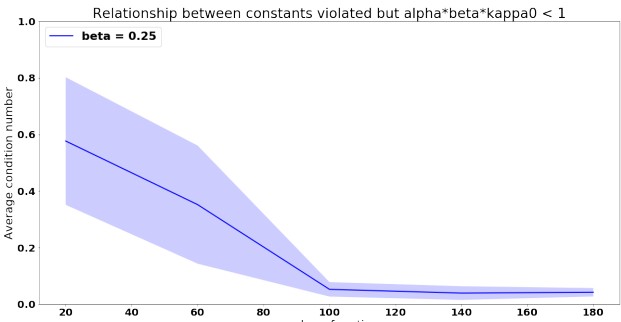

Figure 6: Relationship between the constants violated.

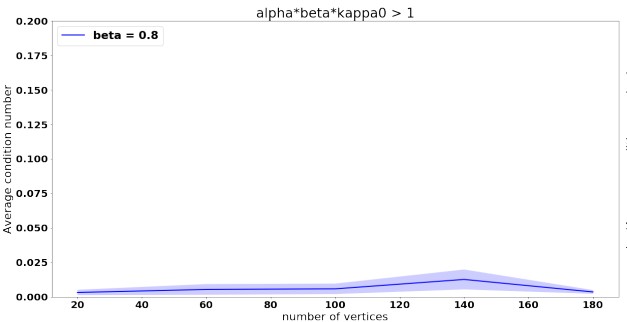

Figure 7: $\alpha * \beta * \kappa_0 > 1$.

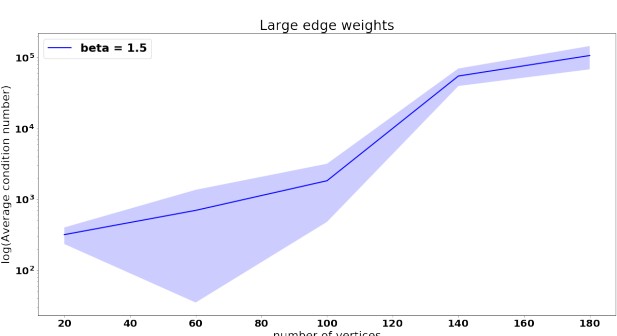

Figure 8: $\beta > 1$ ($y$-axis is in log-scale).

Positive Definite. We compute the covariance matrix from Eq. (2). To compute the condition number, we consider $50$ samples from this model and construct the sample covariance matrix $\tilde{\Sigma}$ which constitutes our perturbed instance. We then compute the average condition number between the exact computation of $\Sigma$ and the one obtained via finite samples. Figure 10, 12, 13 and 11 denotes the results of this experiment. As can be seen, in the sparse regime the condition number is fairly low, while in the dense regime the condition number is almost a factor of $10^2$. Thus, these results indicate two things. First, it verifies the claim in this paper that when the assumptions on range are satisfied the instances are well-conditioned. Second, it also seems to indicate that when range is large, then some of the assumptions in Model 1 are also necessary.

## 6  CONCLUSION

In this paper, we consider the problem of *robust identifiability* in bow-free LSEMs. We give a sufficient condition when bow-free LSEMs can be identified in a robust manner. As a corollary, this implies that all but a tiny set of instances are robustly identifiable. An important open direction is to provide sufficient conditions for robust identifiability in other models of causal inference, particularly the semi-Markovian model (note that Proposition 1.3 in Schulman and Srivastava [2016] is one such sufficient condition). Another is to combine robust identifiability with *model misspecification* (*e.g.,* Cinelli et al. [2019]) where all edges in the model are not correctly specified; Existing works assume access to the *exact* covariance matrix.

## 7 ACKNOWLEDGEMENTS

AL was supported in part by SERB Award ECR/2017/003296 and a Pratiksha Trust Young Investigator Award. AL is also grateful to Microsoft Research for supporting this collaboration.

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
