# OpenReview forum: "Robust Identifiability in Linear Structural Equation Models of Causal Inference"
_auai.org/UAI/2022/Conference — UAI 2022 Poster_

### Official Review · Reviewer_Nka5 · 2022-04-12

**Q2(1) Originality/Novelty:** 2
**Q2(2) Significance/Impact:** 2
**Q2(3) Correctness/Technical Quality:** 3
**Q2(6) Clarity Of Writing:** 2
**Q6 Overall Score:** 5
**Q8 Confidence In Your Score:** 2

**Q1 Summary And Contributions:**

The paper deals with robust identifiability in the sense that small perturbations in the covariance structure only lead to small changes in the edge weights of a DAG. The authors show that for bow-free linear structural equation models the class of robust identifiable models is large.

**Q2 Assessment Of The Paper:**

More detailed information regarding each of these aspects is given below:

**Q2(4) Quality Of Experiments (Optional):**

2: Fair: The experimental evaluation is weak: important baselines are missing, or the results do not adequately support the main claims.

**Q2(5) Reproducibility:**

2: Fair: Key resources (e.g., proofs, code, data) are unavailable but key details (e.g., proof sketches, experimental setup) are sufficiently well-described for an expert to confidently reproduce the main results.

**Q3 Main Strengths:**

The paper has a detailed introduction and preliminaries.

**Q4 Main Weakness:**

I'm not sure about the novelty of the work, the advances over Sankararaman et al. (2019) are there but seem relatively straightforward.

**Q5 Detailed Comments To The Authors:**

In general, I think the formatting of a lot of the math displays and equations could be improved.

Small typo:

-Definition 4 htr instead of htv

**Q7 Justification For Your Score:**

I think the topic and idea of the paper are interesting, but I'm not sure if the advances over existing literature are enough for a new publication.

**Q9 Complying With Reviewing Instructions:**

1: Yes.

---

### Official Review · Reviewer_JwUV · 2022-04-12

**Q2(1) Originality/Novelty:** 2
**Q2(2) Significance/Impact:** 4
**Q2(3) Correctness/Technical Quality:** 3
**Q2(6) Clarity Of Writing:** 3
**Q6 Overall Score:** 6
**Q8 Confidence In Your Score:** 4

**Q1 Summary And Contributions:**

This paper studies the problem of robust identifiability in bow free LSEMs: if the covariance matrix \Sigma was perturbed slightly, the parameters of the graphical model \Lambda also changes slightly. They provided sufficient conditions for bow-free graphs to be robustly identifiable, expanding on the results by Sankararaman et al.

**Q2 Assessment Of The Paper:**

More detailed information regarding each of these aspects is given below:

**Q2(4) Quality Of Experiments (Optional):**

3: Good: The experimental evaluation is adequate, and the results convincingly support the main claims.

**Q2(5) Reproducibility:**

3: Good: Key resources (e.g., proofs, code, data) are available and key details (e.g., proofs, experimental setup) are sufficiently well-described for competent researchers to confidently reproduce the main results.

**Q3 Main Strengths:**

Robust identifiability is an important subclass of identifiability, which will have an important impact on many subfields in AI. This paper provides a graphical criterion for a model to be robust identifiable.

**Q4 Main Weakness:**

The results in this paper is a sufficient condition found by retriving the parameters Lambda from Sigma using a specific algorithm that only works for a subset of graphical models (HTC-identifiable graphs). They do not address the cases for graphs with bows or graphs that are not HTC-identifiable; the parameters in the latter could not be retrieved by the algorithm provided. It may be important to determine if graphs that are not HTC-identifiable are robust indentifiable.

**Q5 Detailed Comments To The Authors:**

- A rigorous definition of robust identifiability should be provided (probably with epsilon-delta)
- Some numerical of graphical models that are (or are not) robustly identifiable may be helpful
- The introduction is too long and should be shortened, some of the paragraphs (such as comments about Foygel's algorithm) could be shifted to later sections. As it stands, the introduction and preliminaries make up over half of the paper length.

**Q7 Justification For Your Score:**

Robust identifiability is currently understudied, but potentially has important implications in AI. However, the results found in this paper are derived from an algorithm that only works on a specific graphical subclass. If the authors could additionally prove that graphs that are not HTC-identifiable are not robust identifiable as well, this paper will appear more complete.

**Q9 Complying With Reviewing Instructions:**

1: Yes.

---

### Official Review · Reviewer_GHya · 2022-04-13

**Q2(1) Originality/Novelty:** 3
**Q2(2) Significance/Impact:** 2
**Q2(3) Correctness/Technical Quality:** 3
**Q2(6) Clarity Of Writing:** 3
**Q6 Overall Score:** 6
**Q8 Confidence In Your Score:** 4

**Q1 Summary And Contributions:**

The paper discusses robust parameter estimation of linear structural equation models. The authors show that bow-free LSEMs are robustly identifiable under some assumptions. The authors show that these assumptions are satisfied with high probability in a specific generative model, and that also in simulated data.

**Q2 Assessment Of The Paper:**

More detailed information regarding each of these aspects is given below:

**Q2(4) Quality Of Experiments (Optional):**

2: Fair: The experimental evaluation is weak: important baselines are missing, or the results do not adequately support the main claims.

**Q2(5) Reproducibility:**

2: Fair: Key resources (e.g., proofs, code, data) are unavailable but key details (e.g., proof sketches, experimental setup) are sufficiently well-described for an expert to confidently reproduce the main results.

**Q3 Main Strengths:**

The paper presents solid theoretical results for robust identifiability of bow-free LSEMs.
The paper is clearly written and easy to follow.
The authors present some intuition on the very technical assumptions.

**Q4 Main Weakness:**

The reasoning behind the random generative model is unclear to me.
The experimental results are limited to examining the effect of various assumptions and graph properties on the condition number.
I am not sure on the novelty compared to Sankararaman 2019.

**Q5 Detailed Comments To The Authors:**

What is the reasoning behind Section 4?  Do we have reason to believe that this generative model is reasonable?
All experimental results are focused on measuring the condition number. I think it would be helpful if some results on the actual errors on the parameter estimation were included.
Figures in the Supplementary (9-13) are cited in the main paper.


**Q7 Justification For Your Score:**

The paper presents solid (as far as I could tell) theoretical results on robust identifiability of LSEMs.

**Q9 Complying With Reviewing Instructions:**

1: Yes.

---

### Official Review · Reviewer_ZPgv · 2022-04-17

**Q2(1) Originality/Novelty:** 3
**Q2(2) Significance/Impact:** 2
**Q2(3) Correctness/Technical Quality:** 3
**Q2(6) Clarity Of Writing:** 3
**Q6 Overall Score:** 5
**Q8 Confidence In Your Score:** 3

**Q1 Summary And Contributions:**

The paper considers the problem of robust identifiability in bow-free LSEMs, as opposed to the more common generic identifiability studied in the literature. The robustness is quantified by a relative condition number and this is bounded under some assumptions. These are tested and illustrated by experiments involving numerical simulations and a gene expression dataset.

**Q2 Assessment Of The Paper:**

More detailed information regarding each of these aspects is given below:

**Q2(4) Quality Of Experiments (Optional):**

2: Fair: The experimental evaluation is weak: important baselines are missing, or the results do not adequately support the main claims.

**Q2(5) Reproducibility:**

2: Fair: Key resources (e.g., proofs, code, data) are unavailable but key details (e.g., proof sketches, experimental setup) are sufficiently well-described for an expert to confidently reproduce the main results.

**Q3 Main Strengths:**

The structure of the paper is clear, and combines both theoretical results and numerical experiments. The result should spark interest in researchers studying robust identifiability of parametric models.

**Q4 Main Weakness:**

All proofs and half of the figures are in the appendix and not in the main text. The code for the experiments is not available.

**Q5 Detailed Comments To The Authors:**

While I am aware it is impossible to fit most content in the main text, one criticism would be that the Appendix has more than twice the number of pages than the main paper (without references). The proof outlines are welcome, but maybe a well placed Lemma would shed more light into the technical aspects of the main result. It is also not great that in the main text one is discussing certain Figures that appear only in the appendix. Conversely, notation that never appears on the main text is introduced, such as that of singular value, eigenvalue, and uniform distribution.

A second point is that reading the abstract emphasizing "we show that robust identifiability is a *strictly* harder problem than generic identifiability" and later on p. 3 "That the access to the exact covariance matrix is not essential under reasonable conditions on the parameters is in fact the main point of our paper", seems to imply that these results are somewhat unexpected. However, as Foygel et al. [2012] show, and the authors heavily use, generic identifiability boils down to "solving a bunch of linear problems", and then, as with the "classical case", one would expect robustness to fail when the relevant matrices are close to being singular and to behave well on a large set away from that locus? Wouldn't at least the abstract sentence be directly implied from the intuition on Assumption 1? To be clear, this is not a criticism diminishing Theorem 1 and 2 (they are precise statements on their own), just about the tone used in presenting such results / general story for the paper.

Finally, the authors mention the work of Ghoshal and Honorio (2018) and say it is not comparable since they severely simplify the structure on the matrix Omega. I slightly disagree: aren't these kinds of models still covered by Theorem 1 (i.e. satisfy the nice conditions of Model 1 easily)? Then (although it wouldn't be a fair comparison), it should still be worth it to compare the sample/computational complexity in that case with the one obtained by the more general setting (e.g. how much better is it?).


Other comments:

- p. 1, Figure 1: the edge from v4 to v6 is missing the arrow marking its direction
- p. 2, after equation (1): "set E puts constraint on" --> "set E puts constraints on"
- p. 3, left, penultimate line: "references therin" --> "references therein"
- p. 4, bottom left: missing the word "value" in "largest singular and eigenvalue"
- p. 4, top right: isn't path length given by the number of edges? After the definition of layer(i) you say layer(1) is the set of vertices without parents, but looks like it should be layer(0) or the definition be adjusted to length i-1.
- p. 4, the notation "spa(v):= pa(pa(v))" is not technically defined since you have explained what pa(v) is for a node (it is a set of nodes) but have not explained what pa(S) for a set of nodes is (presumably the union of the pa(s) for s \in S).
- in Definition 2, is it standard to have sup over "\gamma < 1/(n^4)" ? the exponent "4" appears somewhat arbitrary
- Definition 4: instead of "pa(pa(v))" you can use the introduced notation "spa(v)", also use "\emptyset" instead of "\phi".
- p. 5: maybe you can make clear exactly what assumption in Model 1 makes the noise "adversarial"?
- p. 7, 3rd line Section 5.2: "this explain why" --> "this explains why"
- Simulated dataset: you say "we generate many random graphs exactly as in the main section of the experiments" but when you set k=2 and k=7 you need to make sure that the graphs satisfy the maximum in/out-degree property from Definition 1 (and looks like that's not guaranteed with that scheme)
- p. 8, in the Figures, I find surprising to see the condition number not only staying "stable" when some conditions are violated, but actually *decreasing* in Figure 6 when increasing the number of vertices. How does one explain this? If not exponential increase like in Figure 8, wouldn't one expect a steady (perhaps linear) increase in the other Figures?


**Q7 Justification For Your Score:**

Even in its current version, the listed strengths outweigh the weaknesses and detailed criticisms. Having a good answer / clarification about them would push the paper towards Weak Accept / Accept.

**Q9 Complying With Reviewing Instructions:**

1: Yes.

---

### Decision · Program_Chairs · 2022-05-15

**Decision:**

Accept (Poster)

**Comment:**

Meta Review: This paper studies the question of robust identifiability of the parameters in linear structural equation models. It builds on the work in Sankararaman et al. (2019) and establishes a sufficient condition for a bow-free model to be robustly identifiable (by the algorithm proposed by Foygel et al. 2012), which is shown to hold with high probability under a certain scheme of random generation of the model parameters. The paper also reports a number of experiments with both simulated data and real data.

The reviewers converge on a verdict of "weak/borderline accept", endorsing the importance of the central question and commending the technical quality of the results, but also expressing concerns with the overall novelty and with some presentation issues. In my view, the contributions are sufficiently significant, with nontrivial and interesting results on an important problem that are considerably more general than the current results in the literature. I am therefore inclined to recommend acceptance. And I urge the authors to improve those aspects of the presentation as highlighted in the reviews and address the specific questions about the experimental results raised by reviewer Nka5.